# Provider cost of treating oral potentially malignant disorders and oral cancer in Malaysian public hospitals

Sivaraj Raman [1,2], Asrul Akmal Shafie[1,3]*, Mannil Thomas Abraham[4], Chen Kiong Shim[5], Thaddius Herman Maling[6], Senthilmani Rajendran[7], Sok Ching Cheong[7,8]

1 Discipline of Social and Administrative Pharmacy, School of Pharmaceutical Sciences, Universiti Sains Malaysia, Gelugor, Penang, Malaysia, 2 Pharmacy Department, Hospital Keningau, Ministry of Health, Keningau, Sabah, Malaysia, 3 Institutional Planning and Strategic Center, Universiti Sains Malaysia, Gelugor, Penang, Malaysia, 4 Oral and Maxillofacial Surgery Department, Hospital Tengku Ampuan Rahimah, Ministry of Health, Klang, Selangor, Malaysia, 5 Oral and Maxillofacial Surgery Department, Hospital Umum Sarawak, Ministry of Health, Kuching, Sarawak, Malaysia, 6 Samarahan Divisional Dental Office, Sarawak State Health Department, Ministry of Health, Samarahan, Sarawak, Malaysia, 7 Head and Neck Cancer Research Team, Cancer Research Malaysia, Subang Jaya, Selangor, Malaysia, 8 Department of Oral and Maxillofacial Clinical Sciences, Faculty of Dentistry, University of Malaya, Kuala Lumpur, Malaysia

☯ These authors contributed equally to this work.
* aakmal@usm.my

**Data Availability Statement:** Study data is publicly available in Harvard Dataverse via https://doi.org/10.7910/DVN/2SGZ9T.

## Abstract

Oral cancer has been recognized as a significant challenge to healthcare. In Malaysia, numerous patients frequently present with later stages of cancers to the highly subsidized public healthcare facilities. Such a trend contributes to a substantial social and economic burden. This study aims to determine the cost of treating oral potentially malignant disorders (OPMD) and oral cancer from a public healthcare provider's perspective. Medical records from two tertiary public hospitals were systematically abstracted to identify events and resources consumed retrospectively from August 2019 to January 2020. The cost accrued was used to estimate annual initial and maintenance costs via two different methods- inverse probability weighting (IPW) and unweighted average. A total of 86 OPMD and 148 oral cancer cases were included. The initial phase mean unadjusted cost was USD 2,861 (*SD* = 2,548) in OPMD and USD 38,762 (*SD* = 12,770) for the treatment of cancer. Further annual estimate of initial phase cost based on IPW method for OPMD, early and late-stage cancer was USD 3,561 (*SD* = 4,154), USD 32,530 (*SD* = 12,658) and USD 44,304 (*SD* = 16,240) respectively. Overall cost of late-stage cancer was significantly higher than early-stage by USD 11,740; 95% CI [6,853 to 16,695]; *p* < 0.001. Higher surgical care and personnel cost predominantly contributed to the larger expenditure. In contrast, no significant difference was identified between both cancer stages in the maintenance phase, USD 700; 95% CI [-1,142 to 2,541]; *p* = 0.457. A crude comparison of IPW estimate with unweighted average displayed a significant difference in the initial phase, with the latter being continuously higher across all groups. IPW method was shown to be able to use data more efficiently by adjusting cost according to survival and follow-up. While cost is not a primary consideration in treatment recommendations, our analysis demonstrates the potential economic benefit of investing in preventive medicine and early detection.

**Funding:** The authors would like to acknowledge the Malaysian Ministry of Higher Education for the financial support under the Fundamental Research Grant Scheme (203.PFARMASI.6711655) awarded to Dr. Asrul Akmal Shafie. Sponsor's website: Funder website is accessible from https://mohe.gov.my/en/initiatives-2/187-program-utama/penyelidikan/548-research-grants-information The funders had no role in study design, data collection and analysis, decision to publish, or preparation of the manuscript.

**Competing interests:** The authors have declared that no competing interests exist.

## Introduction

Oral cancer has been recognized as a significant public health crisis, especially in Asia, forming more than half of the global incidence [1, 2]. Out of this, almost 11% were contributed by South-East Asian countries [3]. Despite improvements in diagnosis and therapeutic care, the 5-year survival rates for this region remain lower than 50% due to a combination of reasons, including sociodemographic factors and accretion of risk habits [3]. In Malaysia, oral cancer disproportionately affects the Indian and indigenous groups. Annual national reports also consistently demonstrated a larger proportion of patients are presenting at later stages of tumors, attributing to the substantial disease burden [4, 5].

Malaysia forms an interesting case for universal health coverage in an upper middle-income country. The public healthcare system offers a comprehensive range of health services including cancer treatment, financed mainly through taxation and general revenues from the federal government. While the public health spendings consisted of 43.1% of the total national health expenditures, the sector provided about 75.5% of inpatient care and 64.3% of ambulatory care to the population [6–9]. The fees paid by patients cover both inpatient and outpatient care services, differing by class of accommodation, citizenship, and additional exemptions. However, the charges for Malaysian citizens are heavily subsidized with only 2.6% of expenditures were recovered from patient revenues [8]. Although the commitments for financial risk protection of its population are exemplar, the increasing disease demands and changing demographics continue to put the public health system under strain [10].

Management of oral cancer often involves multiple approaches depending on the cancer stage and patient status. These range from simple surgical resections to multimodal treatment involving radiotherapy and chemotherapy. The introduction of newer diagnostic, pharmacological, and treatment technologies coupled with the long-term care of cancer patients contributes towards a rapid escalation of cost [3, 11]. On top of these, Malaysia's public healthcare system also incorporates additional subsidies for the population above the age of 60 years, which forms most of the oral cancer incidences [12]. Consequently, the bulk of treatment costs will be borne by the Ministry of Health with minimal reimbursement from fee-for-service [7]. The recent implementation of the PeKa B40 scheme, which provides financial incentives for transport and the completion of cancer treatment further adds to this existing financial burden [12].

At the same time, it is well documented that visible abnormal lesions in the oral cavity often precede oral squamous cell carcinomas. These abnormalities, termed as oral potentially malignant disorders (OPMD), have variable chances of transforming to malignant lesions, ranging from 5% to as high as 85% [13]. Given that early intervention such as removing OPMD can halt oral cancer risk, initial screening and treatment can efficiently reduce resource drain. However, albeit being relatively less expensive and less exhaustive, management of OPMD can still lead to substantial financial expenditure from a higher number of cases identified through early screenings.

Analyses of the economic impact of OPMD and oral cancer management to guide decision-makers in Malaysia and its neighboring nations are currently not available [14]. Thus this study aims to provide critical information on their initial and maintenance phase management cost from a public healthcare provider's perspective, stratified by stages. Practical and robust methodological alternatives to estimate annual costs were also explored. This was because longitudinal studies might be a challenge in resource-limited settings, especially when local incidences are low. Findings are expected to be used to consolidate potential savings from early screening and preventive measures. Cost estimates will ultimately allow policymakers to evaluate programs' efficiency and plan for prioritization of resources.

## Materials and methods

The present investigation was carried out as a retrospective activity-based costing study as per the study framework in Fig 1. The study was registered and approved by the Ministry of Health Medical Research Ethics Committee (NMRR -18-3842-45321) and the Universiti Sains Malaysia Human Research Ethics Committee (USM/JEPeM/18120789).

### Population and setting

The study was conducted in Hospital Tengku Ampuan Rahimah (HTAR), Klang and Hospital Umum Sarawak (HUS), Kuching, Malaysia from August 2019 to January 2020. Both are publicly funded tertiary hospitals that are established referral centers for the management of oral cancer and OPMD. They were selected as a study site to capture the diverse patient population in East and West Malaysia.

### Study design and sample

In activity-based costing, costs were assigned according to product and service consumption. The first step was establishing a clinical pathway of care for patients with OPMD and oral cancer. The clinical pathway was developed according to standard treatment guidelines [15], adapted to local practice via consultation with a multidisciplinary team (oral maxillofacial surgeons, oral pathologists, dental public health officers, oncologists, pharmacists and nurses). Fig 2 shows the simplified clinical pathway of a patient journey which was used to guide the costing framework and the construction of data abstraction proforma.

As there was no published information on the cost of oral cancer management locally, the sample size was estimated using available reported data on the cost of colorectal cancer treatment in Malaysia [16]. The calculation was based on detecting a difference of MYR 10,000 (with a standard deviation of MYR 20,000) between early and late-stage cancer, with a power of 80% and a two-sided level of significance of 5% [17]. The sample size obtained was further adjusted according to the case distribution in Malaysia. The National Cancer Registry reported

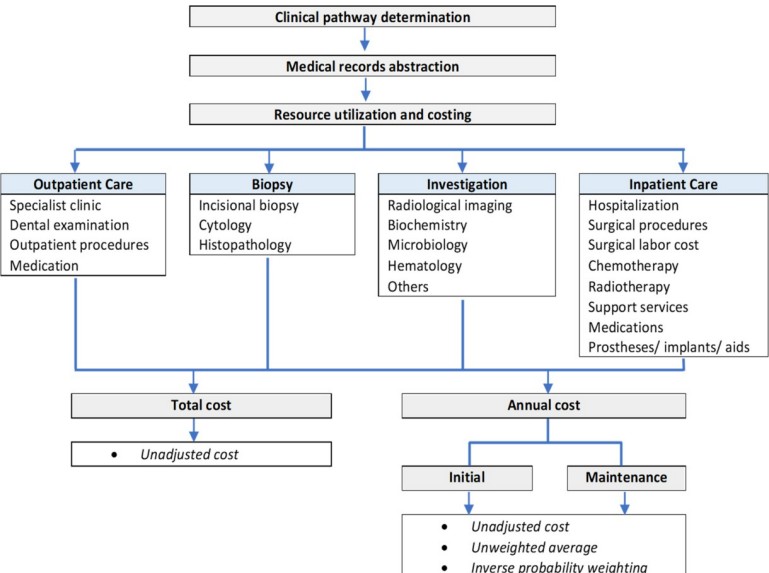

**Fig 1. Study framework to determine the cost of treating OPMD and oral cancer.**

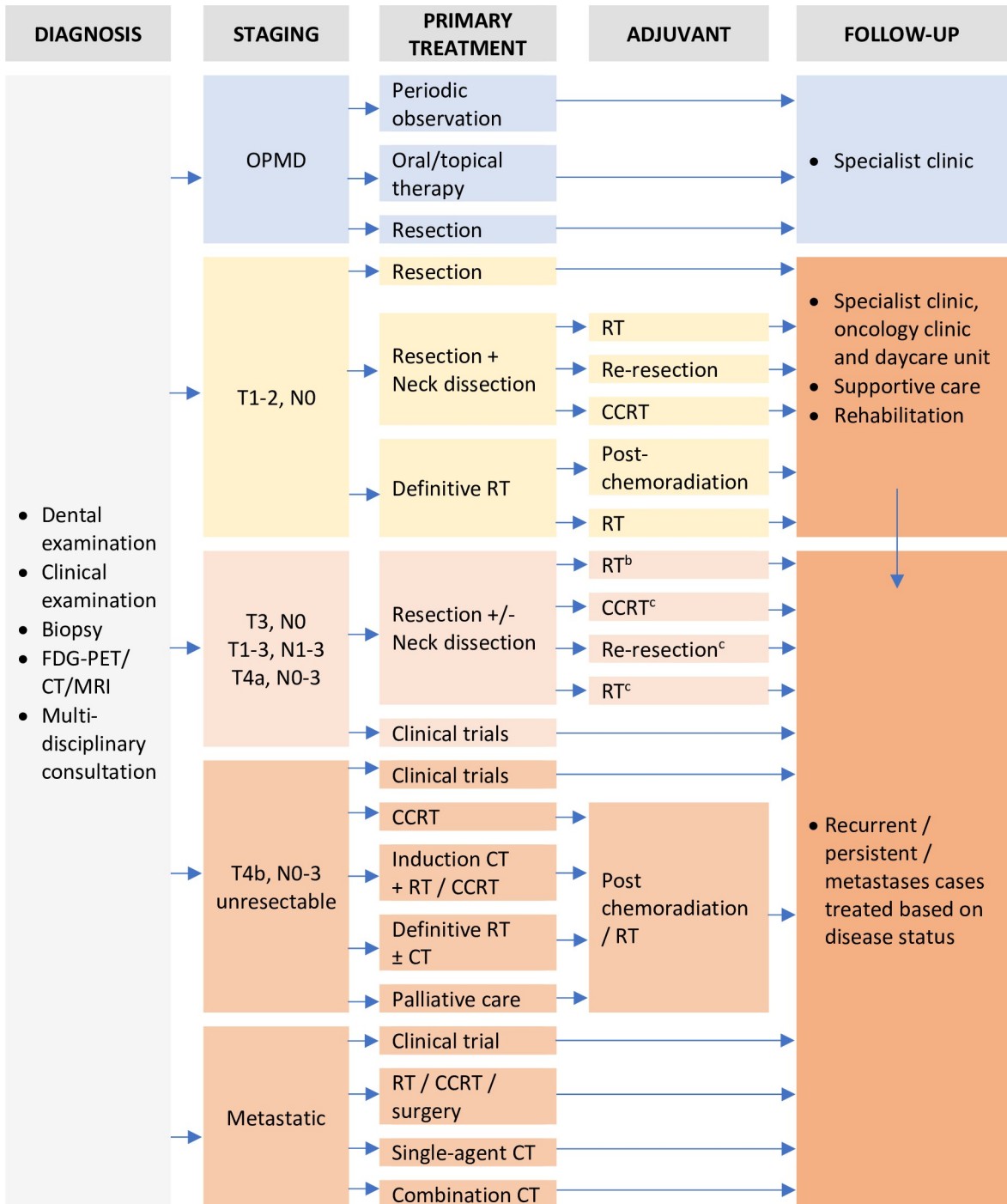

**Fig 2. Simplified clinical pathway of the patient journey in a public healthcare facility.** *Note: The pathway shown is for the treatment of cancer in the buccal mucosa, floor of the mouth, tongue, alveolar ridge, hard palate. [a] FDG-PET = fluorodeoxyglucose-positron emission tomography, CT = computed tomography, MRI = magnetic resonance imaging, RT = radiotherapy, CCRT = concurrent chemotherapy and radiotherapy. [b] Presence (and based on) adverse risk features. [c] No adverse features.

that the number of cases diagnosed at late stages (III and IV) was double that of early stages (I and II) [4]. Accordingly, a total of 80 samples for both OPMD and late-stage cancer in addition to 40 for early-stage cancer was planned.

Patients were identified from the register by stage-stratified convenience sampling and medical records were retrieved. Consent was not required as data were analyzed anonymously. A broad inclusion criterion was set- defined as adult patients above 18, with histologically confirmed oral cancer or OPMD. An oral cancer diagnosis was standardized based on the International Classifications of Diseases 10th revision, consisting of ICD 00 to 06 [18]. The staging of cancer was in accordance with the American Joint Committee on Cancer (AJCC) TNM system. They are based on the extent of the primary tumor (T), regional lymph nodes involvements (N), and the presence of metastasis (M). The information is then combined to a stage grouping and assigned an overall stage based on the guide. The WHO Collaborating Centre for Oral Cancer and Precancer has defined OPMD as a range of lesions or conditions that have a propensity to become malignant. This includes leukoplakia, lichen planus, oral submucous fibrosis, erythroplakia, and other uncategorized histologically-confirmed dysplasias [13].

No minimum duration of follow-up was set for case inclusion. Cases were nevertheless identified for the completion of at least a single treatment. For cancer patients, this involves completion of surgical intervention or chemotherapy or radiotherapy. For OPMD patients, this comprises either surgical procedures or treatment with oral/topical medications or the clinical decision to monitor lesions without any active interventions.

Clinical data collected encompasses specifically the management of OPMD or oral cancer, and any associated events including complications. Investigators only recorded cancer attributable events and services by identifying the primary diagnosis for admission. Events that are less apparent to be distinguished from cancer-related services were omitted. Sociodemographic data were likewise extracted from records and patient registers. The patients' residence was categorized as either 'urban' or 'rural' area according to postcode and township, with an area assignment guide provided by the local government department.

## Valuation and sources

Four components were identified in the costing framework as shown in Fig 1- outpatient, biopsy, investigation, and inpatient. The categorization was made to explore and identify the main cost contributors. Each component's cost was calculated by multiplying the respective elements' frequency with their specified value (S1 Table). Total healthcare costs were generated by summing all four components. Valuations were calculated according to the unsubsidized 'Full-paying non-citizen' tariff in the Medical Fees Order (Cost of Services) 2014. The Malaysian public hospital tariff is grounded on subsidized valuation rather than profit-based [19]. The non-citizen rate was applied as a cost proxy as it best represents unsubsidized charges for clinical services in public healthcare facilities. The values listed were based on private fees survey, consultations with heads of specialty, and available cost information [19]. Although the basis behind the calculation was not reported, a crude comparison of selected dental procedures fees with local cost analysis study validates the cost to be inclusive of capital, operation, maintenance, administrative, and consumables used [20]. The cost of medications was accrued according to prescribing records and based on the 2018 National Medicine Price List obtained from the pharmacy department. All implants, prostheses, and supportive aids were collectively reported as healthcare providers' expenditures to ensure comprehensive costing.

Most of the resource utilization data were from medical records. However, two elements were estimated based on expert opinions. The first element was the human capital cost of extensive multidisciplinary surgical care. Oral cancer surgeries are primarily planned by a minimum of two consultants led teams- one for ablative interventions and another for reconstructive. An average of 8- and 13-hours per operation was assigned for patients undergoing local/regional flap and free flap reconstructive surgery, respectively [21]. For excision of lesions

in OPMD, an average of two hours per surgery was used. This care cost was calculated based on the operating theatre's personnel list and daily remuneration rate.

The second element estimated based on expert opinions was hospitalization and utilization of chemotherapy and radiotherapy. Hospitalization data were not readily available as part of the treatment was conducted at external institutions (National Cancer Institute (NCI) for patients managed in HTAR). However detailed chemotherapy regimens and radiotherapy plans were still obtained from the respective units. Thus, to ensure standardization, hospitalization for each patient was estimated according to the identified individual regimens or plans following standard practice. The minimum period of hospitalization was decided upon discussions with experts consisting of oncologists, radiotherapists, and pharmacists involved in planning chemotherapy regimens in NCI and HUS. For radiotherapy, as regimens were in terms of weeks, five fractions were assumed to equal a week of inpatient admission based on resource use. For chemotherapy, one treatment cycle was estimated as seven days to ease cost calculations. In reality, administration days may range from two to beyond a week following designated regimens or ensuing complications. Both radiotherapy and chemotherapy utilization costs were calculated following standard institutional protocol. Each protocol was itemized and enumerated according to the drugs, additives, and standard monitoring parameters recommended in prescribed regimens obtained from the pharmacy departments.

## Analysis

*Total cost* was generated by tallying all four components identified in the clinical pathway for each patient. This consisted of all healthcare expenses from the first visit until death or last-recorded care event. Annual costs were calculated by applying a phase-specific approach. This was carried out by splitting the total cost according to phases. '*Initial phase*' was set as any period within the 12 months of the first healthcare visit to entail the high early treatment cost. All events and resources consumed after the first year till either death or last care event recorded were documented as '*maintenance phase*' [22, 23]. These formed the preliminary sum of '*unadjusted cost*' which reflected the actual expenditure over various case-time lengths within each respective phase. These values were further used to estimate the mean annual cost using simple averaging as a comparison. This *unweighted average* was calculated by dividing the accrued values in the initial and maintenance phase with their respective time length (in months) and annualized by multiplication with 12.

Broad inclusion criteria were selected to maximize the number of cases recruited and observe cost disparity in real-time. However, one of the drawbacks of such an approach is variability in the follow-up period. Furthermore, a retrospective costing data collection introduces an 'induced informative censoring' from recruitment of patients at different treatment points. Application of standard costing methods with the exclusion of costs from incomplete follow-up will impair the accuracy of inferences. On the other hand, using the traditional survival analysis methods such as the Kaplan-Meier estimator underestimates actual cost from censoring. To overcome such predispositions, an *inverse-probability weighting* (IPW) method by applying Bang and Tsiatis (BT) estimator was adopted [24]. While several different methods are proposed to handle the time-restricted mean cost, the BT estimator was selected due to the ease of single-record-per-subject survival data required relative to other more data-intensive methods.

In the BT estimator, sample weighting was formed using the Kaplan-Meier estimate for censoring $Sc(t_i)$, with the mean IPW total cost estimated as per the equation below. Total costs at each time interval are then multiplied with IPW to adjust for censoring. As uncensored observations are weighted over their inverse probability, cases of early deaths, defaults, or shorter follow-up periods will be weighted less than those followed up till one year [25].

Mean annual cost = $1/N[\sum i Ai(ti)/Sc(ti)]$

N = total sample size of the study, including censored and uncensored patients

$ti$ = time of fixed endpoint, death, or loss to follow up for each patient (in months)

$Ai(ti)$ = the cumulative cost until the time, $t$ for a patient, $i$

$Sc(ti)$ = the probability of being uncensored beyond time, $t$

Categorical variables were presented in frequencies and continuous variables in means (*M*) and standard deviations (*SD*). Kruskal-Wallis, Pearson's Chi-square test, and Fisher's exact test were used to explore differences between sociodemographic factors and enlisted cases. Comparison between OPMD and oral cancer stages was investigated using the Kruskal-Wallis test. In addition, the calculated means and standard deviations were used to compare crude differences among annual costs between all three methods applied. The assumption made for the comparative analysis was that the sample was large enough to apply a central limit theorem [24].

While cost data often do not conform to parametric assumptions, the estimates were described in means to ensure robustness and pragmatism in guiding decisions [26]. Such reporting was in line with good research practice guidelines for cost analysis by the Professional Society for Health Economics and Outcomes Research (ISPOR). Costs were reported in both Malaysian Ringgit (MYR) and US Dollars (USD) without year adjustment. The conversion rate was based on the 2019 purchasing power parity (PPP) to consider Malaysia's economic productivity and standards (1 USD = 1.602 MYR). All analyses were conducted using Stata version 14.0 (StataCorp, College Station, Texas 77845 USA).

## Results

### Sociodemographic characteristics

A total of 86 OPMD and 148 oral cancer cases were included in the study. The number of cases according to TNM staging were: Stage I (n = 18, 12.2%); Stage II (n = 24, 16.2%); Stage III (n = 31, 20.9%) and Stage IV (n = 75, 50.7%). For the types of OPMD, lichen planus was the most common (n = 45, 52.3%), followed by leukoplakia (n = 12, 14.0%), oral submucous fibrosis (n = 3, 3.5%), mixed red/white lesions (n = 2, 2.3%) and others (n = 24, 27.9%). The '*others*' proportion consisted largely of lesions recorded as dysplasia.

Table 1 shows the sociodemographic characteristics and clinical details of the subjects. There was a significant difference in ethnicity between both OPMD and cancer groups, with Indians prevailing in both OPMD and late-cancer stages. The apparent difference in education level and occupation between groups might be partly contributed by disproportional missing data. Buccal mucosa remained the predominant site for both OPMD and late cancer. More than half of early-stage cancer detected were primarily located in the tongue. A disparity in TNM staging among oral cancer patients was observed in both facilities, with Stage III and Stage IV forming around 71.6% of total cases, reflecting the proportion reported in the National Cancer Registry.

### Healthcare cost and consumption

Overall, the total cost of managing disease increased with severity, from an average of MYR 6,631; *SD* = 7,113; 95% CI [5,120 to 8,142] (*M* = USD 4,139; *SD* = 4,400; 95% CI [3,196 to 5,082]) in OPMD to MYR 56,820; *SD* = 17,529; 95% CI [51,491 to 62,149] (*M* = USD 35,468; *SD* = 10,942; 95% CI [32,142 to 38,795]) and MYR 71,536; *SD* = 24,047; 95% CI [66,935 to 76,138] (*M* = USD 44,654; *SD* = 15,011; 95% CI [41,782 to 47,527]) in early and late-stage cancer respectively. Table 2 shows the detailed breakdown of initial phase cost per patient by care components and diagnosis. The initial phase mean unadjusted cost was MYR 4,583;

**Table 1. Sociodemographic and clinical details of patients.**

| Characteristic | | OPMD (n = 86) | Early Cancer (n = 42) | Late Cancer (n = 106) | p-value[a] | |
|---|---|---|---|---|---|---|
| | | *Mean (SD)* | *Mean (SD)* | *Mean (SD)* | *All group* | *Cancer* |
| **Follow-up duration (months)** | Initial | 8.5 (4.4) | 9.7 (3.8) | 8.6 (3.8) | 0.447 | 0.219 |
| | Maintenance | 47.0 (46.4) | 60.8 (46.0) | 41.9 (37.9) | 0.208 | 0.081 |
| **Age** | | 60.2 | 59.9 | 61.6 | 0.802 | 0.542 |
| | | *Freq (%)* | *Freq (%)* | *Freq (%)* | *All group* | *Cancer* |
| **Gender** | Male | 32 (37.2) | 24 (57.1) | 50 (47.2) | 0.091 | 0.274 |
| | Female | 54 (62.8) | 18 (42.9) | 56 (52.8) | | |
| **Ethnicity** | Malay | 21 (24.4) | 8 (19.1) | 20 (18.9) | 0.024 | 0.294 |
| | Chinese | 17 (19.8) | 14 (33.3) | 28 (26.4) | | |
| | Indian | 41 (47.7) | 9 (21.4) | 39 (36.8) | | |
| | Indigenous | 7 (8.1) | 11 (26.2) | 19 (17.9) | | |
| **Location** | Urban | 50 (58.1) | 26 (61.9) | 57 (53.8) | 0.636 | 0.369 |
| | Rural | 36 (41.9) | 16 (38.1) | 49 (46.2) | | |
| **Education** | None | 7 (8.1) | 3 (7.1) | 10 (9.4) | 0.033 | 0.062 |
| | Primary | 12 (14.0) | 1 (2.4) | 23 (21.7) | | |
| | Secondary or higher | 36 (41.9) | 13 (30.9) | 25 (23.6) | | |
| | Not available[b] | 31 (36.1) | 25 (59.5) | 48 (45.3) | | |
| **Occupation** | Not working | 23 (26.7) | 2 (4.8) | 27 (25.5) | 0.024 | 0.007 |
| | Employed | 23 (26.7) | 18 (42.9) | 22 (20.8) | | |
| | Retired | 14 (16.3) | 5 (11.9) | 14 (13.2) | | |
| | Not available[b] | 26 (30.2) | 17 (40.5) | 45 (42.5) | | |
| **Anatomic site** | Buccal mucosa | 54 (62.8) | 8 (19.1) | 39 (36.8) | <0.001 | 0.006 |
| | Tongue | 20 (23.3) | 25 (59.5) | 31 (29.3) | | |
| | Alveolar | 2 (2.3) | 2 (4.8) | 14 (13.2) | | |
| | Others[c] | 10 (11.6) | 7 (16.7) | 22 (20.8) | | |

'*All group*' difference was compared between OPMD, early- and late-cancer, while '*cancer*' comparison was between early- and late-cancer alone.

[a] Kruskal-Wallis H test with significance set to $p < 0.05$ was applied on continuous variables and Pearson Chi-square test for proportions with significance set to $p < 0.05$ for categorical variables.

[b] Data not available in medical records.

[c] Consists of the gingiva, lip, floor of mouth, palate, mandible, and other sites.

**Table 2. Unadjusted healthcare provider cost per patient in the initial phase (in MYR).**

| | OPMD | | Early cancer | | Late cancer | | p-value[a] | |
|---|---|---|---|---|---|---|---|---|
| | *Mean* | *SD* | *Mean* | *SD* | *Mean* | *SD* | *All group* | *Cancer* |
| Outpatient | 1,632 | 853 | 2,527 | 1,211 | 2,648 | 1,322 | <0.001 | 0.763 |
| Biopsy | 1,280 | 702 | 1,367 | 646 | 1,417 | 765 | 0.667 | 0.958 |
| Investigation | 456 | 843 | 5,036 | 2,005 | 6,255 | 2,360 | <0.001 | 0.029 |
| Inpatient | 1,216 | 3,135 | 41,391 | 16,356 | 56,443 | 19,244 | <0.001 | <0.001 |
| **Total** | 4,583 | 4,082 | 50,321 | 16,053 | 66,762 | 20,195 | <0.001 | <0.001 |

'All group' difference was compared between OPMD, early- and late-cancer, while 'cancer' comparison was between early- and late-cancer alone.

[a] Kruskal-Wallis H test with significance set to p < 0.05.

*SD* = 4,082; 95% CI [3,716 to 5,451] (*M* = USD 2,861; *SD* = 2,548; 95% CI [2,320 to 3,403]) in OPMD and MYR 62,097; *SD* = 20,458; 95% CI [58,783 to 65,409] (*M* = USD 38,762; *SD* = 12,770; 95% CI [36,694 to 40,830]) for the treatment of cancer. The unadjusted cost showed that only expenditures for biopsy remained equivalent across all three groups, while the rest showed significant differences. It was evident that the large difference in inpatient expenses ultimately drove the higher overall cost of managing late-stage cancer compared to early-stage.

Inpatient care contributed to 47.8% of the overall total cost to treat cancer. However, in the initial phase, this formed a staggering 84.0% of the expenses. A further breakdown of inpatient care in Fig 3 showed that the leading cost drivers were surgical interventions and radiotherapy, followed by hospitalization, personnel, and chemotherapy. Other elements, such as prostheses, medications, and support services such as rehabilitation and occupational therapy, were comparatively small. On the other hand, the outpatient component remained the major cost contributor in OPMD management.

The cost of individual treatment over the follow-up period was on average MYR 28,642; *SD* = 7,660; 95% CI [27,145 to 30,139] (*M* = USD 17,879; *SD* = 4,782; 95% CI [16,944 to

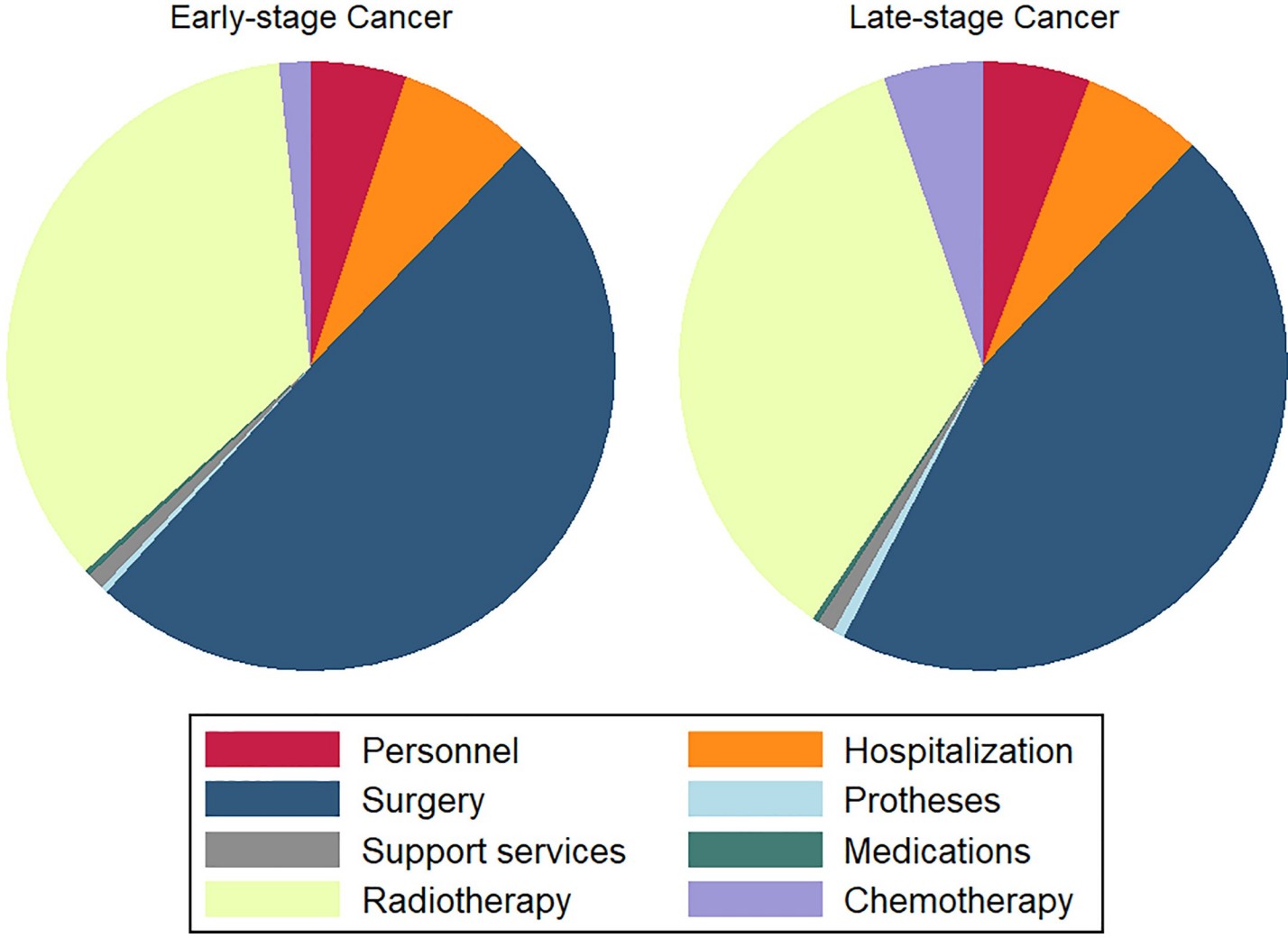

**Fig 3. Chart of inpatient cost breakdown for early- and late-stage cancer.**

**Table 3. Total cost per patient, stratified by treatment modalities (in MYR).**

| Treatment modalities | | n | *Mean* | *SD* | *95% CI* | *Min* | *Max* |
|---|---|---|---|---|---|---|---|
| **OPMD** | Observation | 25 | 3,000 | 1,928 | 2,204–3,795 | 1,928 | 8,375 |
| | Oral/topical | 45 | 5,363 | 4,282 | 4,076–6,650 | 1,916 | 19,637 |
| | S | 16 | *15,872* | *10,443* | 10,307–21,437 | 4,458 | 40,188 |
| **Early-stage cancer** | R | 1 | 32,532 | - | - | - | - |
| | S | 19 | 44,397 | 12,235 | 52,625–69,332 | 24,767 | 69,582 |
| | S + R | 15 | 64,925 | 11,560 | 58,522–71,327 | 51,183 | 89,779 |
| | S + CCRT | 7 | 76,639 | 11,369 | 66,125–87,154 | 60,170 | 89,866 |
| **Late-stage cancer** | R | 8 | 45,684 | 15,832 | 32,448–58,921 | 18,276 | 60,955 |
| | CCRT | 10 | 51,769 | 11,206 | 43,753–59,785 | 33,963 | 68,041 |
| | S | 26 | 60,978 | 20,681 | 52,625–69,332 | 29,859 | 112,044 |
| | S + C | 7 | 71,970 | 12,157 | 60,727–83,214 | 52,328 | 85,364 |
| | S + R | 22 | 77,220 | 20,478 | 68,141–86,300 | 41,817 | 145,220 |
| | S + CCRT | 33 | 88,231 | 21,925 | 80,457–96,005 | 33,587 | 144,913 |

S = surgery, R = radiotherapy, C = chemotherapy, CCRT = concurrent chemotherapy and radiotherapy, CI = confidence interval.

18,813]) for radiotherapy, MYR 5,794; *SD* = 5,150; 95% CI [4,428 to 7,161] (*M* = USD 3,617; *SD* = 3,617; 95% CI [2,764 to 4,470]) for chemotherapy and MYR 23,649; *SD* = 12,248; 95% CI [21,777 to 25,520] (*M* = USD 14,762; *SD* = 7,645; 95% CI [13,594 to 15,930]) for surgical procedures. Both radiotherapy and chemotherapy costs showed no significant differences between early- and late-stage cancers. Although there was a wide range of chemotherapy regimens adopted based on disease status and patient response, the most common protocols prescribed consisted of Cisplatin, Carboplatin, 5-Fluorouracil, Docetaxel, and Gemcitabine. The average surgical procedures cost, however, was significantly higher in later stages (provided in S2 Table). It was predominantly contributed by the complexity of the procedures, especially reconstructive surgeries. The mean cost of prostheses and implants was MYR 1,255; *SD* = 1,026; 95% CI [897 to 1,549] (*M* = USD 783; *SD* = 640; 95% CI [560 to 967]). This was an over-estimation as implants for reconstruction, in reality, would be largely borne by patients. Healthcare providers will only subsidize the standard supportive aids.

Treatment modalities and their respective total management costs were further analyzed and reported in Table 3. Most OPMD patients were treated with medications such as oral and topical corticosteroids, retinoids, or peripheral vasodilators such as pentoxifylline. Surgical excision of lesions was conducted in 18.6% of cases. The rest were continued to be monitored closely with lifestyle and risk factor modification in specialist clinics. In contrast, the core treatment modality for patients with cancer was surgery, either as a primary treatment or in combination with other therapies. As expected, treatment modality and type exhibited an association with cancer stages. It was generally anticipated that the proportion of multimodal therapies to be larger in late stages compared to the early stages, in line with treatment recommendations. Intriguingly there was no significant difference between the proportion of early and late cancer patients needing multimodality treatment in our sample, at 62.0% versus 52.4%, $X^2$ (1, n = 163) = 1.193; *p* = 0.275, respectively.

Table 3 also illustrates the potential range of expenditures required to treat a patient in a public healthcare setting. Multimodal treatment consistently incurred higher expenditures relative to monotherapies at all stages. Large variations in the minimum and the maximum values reflected the diverse follow-up length and complexity of management of included cases.

**Table 4. Annual cost estimate per patient (in MYR).**

| | OPMD | | Early cancer | | Late cancer | | *p*-value[a] | |
|---|---|---|---|---|---|---|---|---|
| | *Mean* | *SD* | *Mean* | *SD* | *Mean* | *SD* | **All group** | **Cancer** |
| **Unweighted average**[b] | | | | | | | | |
| Initial | 9,448 | 9,251 | 84,666 | 64,947 | 121,583 | 80,766 | <0.001 | <0.001 |
| Maintenance | 1,305 | 1,121 | 5,931 | 12,532 | 5,833 | 9,851 | <0.001 | 0.108 |
| **Inverse probability weighting**[c] | | | | | | | | |
| Initial | 5,705* | 6,655 | 52,113* | 20,278 | 70,975* | 26,017 | <0.001 | <0.001 |
| Maintenance | 1,383 | 1,555 | 2,843 | 6,395 | 3,963 | 5,193 | 0.003 | 0.457 |

[a] Kruskal-Wallis H test with significance set to $p < 0.05$.

[b] Accrued cost is divided by the time length (in months) and transformed to 1-year value by multiplication with 12 months.

[c] Cost estimation using Bang and Tsiatis estimator, with the difference between means measured using independent T-test.

*significantly different when compared with unweighted average cost based on an independent T-test with significance set to $p < 0.05$.

### Annual cost estimation

Annual cost estimated using unweighted average and IPW method was reported in Table 4. Both estimates were consistent with unadjusted cost, demonstrating a significant difference in the initial phase between cancer groups. A crude comparison of IPW estimate with unweighted average showed a significant difference in the initial phase cost, with the latter being continuously higher across all groups. In contrast, there was no significant difference in the IPW values compared to unadjusted costs. The calculated average mean difference of late-stage cancer cost with early-stage cancer in the initial phase was MYR 18,862; *SE* = 4,022; 95% CI [10,979 to 26,746] (*M* = USD 11,774; *SE* = 2,511; 95% CI [6,853 to 16,695]) and between OPMD and cancer was MYR 58,818; *SE* = 2,330; 95% CI [54,252 to 63,385] (*M* = USD 36,715; *SE* = 1,454; 95% CI [33,865 to 39,566]) based on the reported IPW values (S3 Table). In contrast, no significant difference was identified between both cancer stages in the maintenance phase, MYR 1,120; 95% CI [-1,829 to 4,070]; *p* = 0.457 (USD 700; 95% CI [-1,142 to 2,541]).

## Discussion

### Cost difference and drivers in oral cancer

Our average cost of treatment of MYR 52,113 (USD 32,530) in early-stage and MYR 70,975 (USD 44,304) in late-stage cancer was shown to be comparable to the findings by Epstein et al. in California, based on Medicaid reimbursement data inflated to the year 2019 [27]. However, it was predominantly lower than the rest of the studies conducted in developed nations but higher than those in developing countries such as India, Sri Lanka, and Iran (S4 Table) [28–36]. The comparison showed a clear distinction in healthcare tariffs between countries. Low and middle-income nations relatively spent lesser per patient due to multiple factors such as availability of state-of-the-art therapies and technologies, accessibility to newer chemotherapeutic agents, and overall cost of living. Variations between studies were likewise attributable to the costing approach and incorporated components.

Locally the costing studies are limited to a handful of cancers and are often based on newer chemotherapeutic agents alone. An extensive costing study conducted in a similar setting in Malaysia by Azzani et al. among colorectal cancer patients was a good parallel example for comparison. They reported the first-year cost to treat a patient ranged from USD 4,410 in Stage I to USD 9,023 in Stage IV [16]. These values were distinctly lower than our findings albeit comprising of comparable cost components. The difference was predominantly

contributed by both the costing and resources consumed. Oral cancer in our study incurred a larger cost for both oncological and surgical interventions. This was partly because we were able to capture a more comprehensive cost associated with surgeries and concurrent chemotherapy and radiotherapy in oral cancer especially in terms of complications. Consistently Jacobson et al. also reported that the medical cost of oropharyngeal cancer was generally higher than other cancers [29].

Expenditures in the late stages of oral cancer in our setting were chiefly inflated by higher surgical intervention and the associated labor cost. Patients in later stages required more extensive excisions coupled with more complex reconstructive surgeries. For example, almost half of surgeries in later stages involved resource-intense microvascular free flap reconstructions, which cost MYR 15,454 (USD 9,647) per surgery. On the contrary, some of the patients in Stage I and II only required primary excisions with general reconstructive surgeries, which costs MYR 3,000 (USD 1,873). Subsequently, the labor-intense surgical procedures also consumed more staff and time from the multidisciplinary team.

Treating Stage III and IV cancers were more expensive in the initial phase and gave rise to higher expenditure over the total management period relative to early-stage cancer and OPMD. Our results notably echoed the findings from numerous studies worldwide illustrating the higher economic burden of treating late-stage oral cancer [28, 33, 35, 37]. The global trend was generally due to a higher percentage of multimodal cancer treatment and extended hospitalization in later stages. However, there was no noteworthy difference in both the proportion of multimodal treatment and hospitalization cost between both cancer groups in our data.

Our sample exhibited a different distribution pattern of patients to such studies. For example, a similar study by Zavras et al. among oral squamous cell carcinoma patients in Greece reported a significant association between treatment modality and staging. In their study, 90% of Stage I patients were treated with only surgery, while more than 90% in advanced stages underwent multimodal treatment [33]. In comparison, our study reported a value of 47.6% and 62.0% respectively. The unconventionality in our study is likely to have resulted from our inclusion criteria and sampling approach.

We included all patients with at least a single completion of treatment modality. Consequently, many cases in later stages may have only completed a part of their planned combination therapy within the data collection period. Thus our data presented a larger proportion of monotherapies amongst late-stage patients than expected. In terms of sampling, half of the patients in the early stages were treated with a multimodal approach. While differing from other studies, the cases were still managed according to treatment guidelines [15]. Almost 60% of early-stage cancer were in the tongue with more than half requiring radiotherapy or concurrent chemotherapy and radiotherapy. The larger percentage may have occurred from the sampling of more complicated cases involving adverse risk features such as positive margins and invasions in the early stages.

## Divergences and accuracy of cost estimates

The selection of analytical methods to generate mean healthcare cost is critical, especially to project exact and precise values. The standard in cost analysis will be the enumeration of a sufficiently powered sample size over one year [23, 28, 35]. Such practice provides an accurate annual cost and distribution while avoiding the difficulty of censored cases. Nevertheless, this consumes a vast amount of resources to recruit a fair number of samples and losses the wealth of data from patients with poorer prognosis or adherence. The alternative approach was to allow for more flexible inclusion criteria by compensating the variability with vigorous estimation methods. In our study at local referral centers, the latter was more appealing as the

prevalence of oral cancer cases was relatively lower at early stages, alongside the wide-ranging complexity and compliances to treatment [25]. Thus our values better represent the spectrum of the reality of cost incurred in public hospitals for planning and projections.

The initial phase expenditure between all three methods demonstrates the strength and weaknesses of each approach. For unadjusted costing, incomplete cases and shorter follow-ups prevent the potential expenses from being established. The unequal distribution of values between groups also reduces the robustness of analysis to identify meaningful differences in smaller samples. On the other hand, unweighted averaging causes overestimation of cost. There is a tendency for cases with shorter durations to be inflated during the annualization process. Such limitations are imperative in cancer because costs are often concentrated in the first few months. In contrast, the IPW method allowed existing data to be used more efficiently by adjusting costs according to survival and follow-up. This approach controls both overestimations and generates asymptotically normal distribution [24, 25].

### Economic burden and recommendations

Cost reduction strategies in oral cancer principally should aim to decrease the need for complex reconstructive surgeries and procedures. This is only achievable if the tumor does not extend or invade beyond the primary site. Detecting oral cancer at the earlier stages can significantly reduce the average healthcare cost per patient by MYR 18,862 (USD 11,740). Furthermore, precancer screening and prevention of malignant transformation should also be emphasized. While the proportion of cases may increase, this can be offset by the overall resource-saving. For every case detected earlier at the OPMD stage and prevented from progressing, the state saves an average of MYR 58,818 (USD 36,715) per patient in the first year. Dividing this difference in cost with the initial phase expenditure of treating OPMD corresponds to being able to treat ten OPMD patients.

The actual public healthcare burden of oral cancer is likely to be higher than estimated in this study. Due to the nature of the study being retrospective and lacks centralized record-keeping, many potential indirect hospitalizations from oral cancer may not be included in the costing. Concurrently, the lack of longitudinal costing for both chemotherapy and radiotherapy besides approximation based on standard protocols may underestimate the actual cost burden incurred. Electronic case recording, coupled with a case-mix system's implementation, may be a valuable tool for future analysis and projections. Secondly, using hospital tariffs grossly underestimates the expenditures from fixed cost components such as shared facilities and other indirect health consequences besides elements like administrative costs as the basis for the fees list is not fully characterized. Lastly, and more importantly, the economic burden is expected to increase multifold if patient cost and loss of productivity from the disease were included [36, 37]. Thus, the overall value of preventions may be enormous and staggering than predicted in this study.

### Conclusion

This study was the first data analysis and exploration into the cost of managing OPMD and oral cancer in Malaysia under a subsidized public healthcare system. It showed a significant cost burden in treating patients with late-stage cancer and multimodal therapy. Additionally, our in-depth comparison between estimate methods provided an insight into the importance of selecting a suitable analytical approach. These methods offer a robust alternative in cost analysis in settings where sample size may suffer from strict inclusion criteria or when study resources are limited.

Our research validates the potential economic benefit of investing in preventive medicine in oral cancer as the financial and social commitment is expected to continue in an upward

trend. This is because as newer therapies and financial incentives are introduced for patients with cancer, in addition to the risks of an aging population, the overall expenditures sustained by the ministry inflates. We hope that our values might guide decision-makers in the prioritization of resources and the development of control plans.

## Supporting information

**S1 Table. Healthcare resource utilization frameset.**
(PDF)

**S2 Table. Inpatient factors impacting cost difference.**
(PDF)

**S3 Table. Annual cost estimate and difference based on IPW method.**
(PDF)

**S4 Table. Literature review of oral cancer management cost worldwide.**
(PDF)

## Acknowledgments

We would like to thank the Hospital Directors, and Oral and Maxillofacial Surgery Departments of Hospital Tengku Ampuan Rahimah and Hospital Umum Sarawak for their tremendous support. We extend our deepest gratitude to Mdm Margareth Cheliah, Dr. Deeban Dass, Mr. Eric Lee and Ms. Jackyln Gopal who facilitated the data collection. We also express our thanks to the Director-General of Health Malaysia for his permission to publish this paper.

## Author Contributions

**Conceptualization:** Asrul Akmal Shafie, Sok Ching Cheong.

**Formal analysis:** Sivaraj Raman.

**Funding acquisition:** Asrul Akmal Shafie.

**Investigation:** Sivaraj Raman, Senthilmani Rajendran, Sok Ching Cheong.

**Methodology:** Sivaraj Raman, Asrul Akmal Shafie, Senthilmani Rajendran, Sok Ching Cheong.

**Project administration:** Sivaraj Raman, Asrul Akmal Shafie, Mannil Thomas Abraham, Chen Kiong Shim, Senthilmani Rajendran, Sok Ching Cheong.

**Resources:** Asrul Akmal Shafie.

**Supervision:** Asrul Akmal Shafie, Sok Ching Cheong.

**Validation:** Asrul Akmal Shafie, Mannil Thomas Abraham, Chen Kiong Shim, Thaddius Herman Maling, Sok Ching Cheong.

**Writing – original draft:** Sivaraj Raman.

**Writing – review & editing:** Sivaraj Raman, Asrul Akmal Shafie, Mannil Thomas Abraham, Chen Kiong Shim, Thaddius Herman Maling, Sok Ching Cheong.

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
