## [Decision Letter · Decision Letter 0]

26 Mar 2021

PONE-D-21-05143

Cost of Treating Oral Cancer in Malaysia: A Public Healthcare Perspective

PLOS ONE

Dear Dr. SHAFIE,

Thank you for submitting your manuscript to PLOS ONE. After careful consideration, we feel that it has merit but does not fully meet PLOS ONE’s publication criteria as it currently stands. Therefore, we invite you to submit a revised version of the manuscript that addresses the points raised during the review process.

The reviewers have suggested revisions to strengthen the paper. Please address these. In regard to the comment requesting more information about the Malaysian health system, you do not need to comment extensively, but to focus on who pays for what in the treatment, and how that affects how you have measured the costs.

We look forward to receiving your revised manuscript.

Kind regards,

Susan Horton

Academic Editor

PLOS ONE

Journal Requirements:

Reviewers' comments:

Reviewer's Responses to Questions

**Comments to the Author**

1. Is the manuscript technically sound, and do the data support the conclusions?

Reviewer #1: Partly

Reviewer #2: Yes

2. Has the statistical analysis been performed appropriately and rigorously? 

Reviewer #1: Yes

Reviewer #2: Yes

3. Have the authors made all data underlying the findings in their manuscript fully available?

Reviewer #1: No

Reviewer #2: Yes

4. Is the manuscript presented in an intelligible fashion and written in standard English?

Reviewer #1: Yes

Reviewer #2: Yes

5. Review Comments to the Author

Reviewer #1: The article attempts to determine the costs of treating oral potentially malignant disorders (OPMD) and oral cancer in Malaysia from a healthcare provider’s perspective. The authors have aimed to fill an important missing gap in the current literature on the cost of OPMD and oral cancer in Malaysia and also to provide policy recommendation on the need for early detection and prevention.

Besides trying to fill an important gap in literature, the study includes both East and West Malaysia resources use which is one of the key strengths of the study as well as the comprehensiveness of the study.

However, the study has several limitations which may need to be addressed:

Time of study period is short (6 months):

• To do a comprehensive costing, time-period should be longer to capture follow-up treatment of cancer and resources use. Explain why this time period is used and whether the study could be extended for a longer time period.

Background

• Provide background of health system in Malaysia and whether all the costs determined in the analysis are covered by the government. Are there any out-of-pocket costs or costs from private insurers that are not included?

Costing seems incomplete.

• Since the study did not fully include one of the key drivers of costs based on external institutions data (hospitalization and utilization of chemo) and instead relied on expert opinion (perhaps one 1 expert), the costing is not fully accurate. To do a full costing, attempts should be made to get data from the National Cancer Institute and other external institutes to obtain the resource use of hospitalizations and chemo treatment.

• Provide more details on the constitution of the inter-disciplinary team – which institutes etc.? Any reason a hospital pharmacist was not involved who might be able to provide more cost details and treatment pathways?

• More information required on the types of chemo drugs that were considered as part of the costing.

• Also needs to show the variation of costs for different patients– may need some form of probability distribution function to show the variation in costs is recommended.

• Consider inclusion of process map that shows the journey of the patient through the healthcare system

• The title only indicates oral cancer when the study includes both oral cancer and OPMD costing.

Method

• More rationale provided on the IPW method and why the authors think this costing method is the most appropriate for undertaking this study and how the costs were adjusted for follow-up and survival. Have the authors considered other methods and if so, why are those methods excluded from the study? How does the study handle uncertainty in costs and the skewed nature of healthcare costs?

• What constitute maintenance costs?

• The study did not seem to include any administrative costs in the total costs (e.g. utilities, personnel, surgical overhead costs). These may need to be determined or approximated so that costs provided are not under-estimated.

• Inclusion of sensitivity analysis to show variation in costs

Minor Edit

• Suggest to change the word ‘manpower’ to personnel, workforce or professional services etc.

Reviewer #2: Authors have attempted to investigate important area of research. Economic cost assessments are very important to eye open the policy makers. Following comments to improve the manuscript for publication.

1 Generally, costing studies describes direct and indirect cost. However, this study focuses only on direct health care cost which is part of the costing analysis.

2. Term “oral cancer”, generally exclude salivary gland tumour, what is the specific reason to include the ICD C7 and C8.

3 Is the follow up visit cost taken into considerations.

4. Generally OPMD patients are treated in the Out Patient Department, however, considerable amount money spent on inpatient segment in table 2.

5. In cost calculation, valuation was based on the “full paying patient”, according to my understanding, is it subsidized valuation or profit based valuation. It is not clear to the reader.

6. PLOS authors have the option to publish the peer review history of their article (what does this mean?). If published, this will include your full peer review and any attached files.

Reviewer #1: **Yes: **Mayvis Rebeira

Reviewer #2: No

---

## [Author Response · Author response to Decision Letter 0]

17 Apr 2021

Thank you for submitting your manuscript to PLOS ONE. After careful consideration, we feel that it has merit but does not fully meet PLOS ONE’s publication criteria as it currently stands. Therefore, we invite you to submit a revised version of the manuscript that addresses the points raised during the review process.

The reviewers have suggested revisions to strengthen the paper. Please address these. In regard to the comment requesting more information about the Malaysian health system, you do not need to comment extensively, but to focus on who pays for what in the treatment, and how that affects how you have measured the costs

• Thank you for the suggestion. We have described universal health coverage in Malaysia in the Introduction section. We have also highlighted the public healthcare system which offers a comprehensive range of health services including cancer treatment, financed mainly through taxation and general revenues from the federal government [page 2, line 70-72]. 

• The fees for medical services are gazetted under the Fees Act 1951. The fees paid by patients cover both inpatient and outpatient care services, differing by class of accommodation, citizenship, and additional exemptions. However, the charged fees for Malaysian citizens are heavily subsidized with only 2.6% of expenditures were recovered from patient revenues [page 2, line 75-77]. 

• On the other hand, non-citizens are required to pay for an unsubsidized rate. Thus the ‘Full-paying non-citizen’ tariff in the Medical Fees Order (Cost of Services) 2014 was used as a cost proxy as it best represents unsubsidized charges for clinical services in public healthcare facilities. The values were based on private fees survey, consultations with heads of specialty, and available cost information. Although the basis behind the calculation was not reported, a crude comparison of selected dental procedures fees with local cost analysis study validates the cost to be inclusive of capital, operation, maintenance, administrative, and consumables used [page 6, line 187-192].

If applicable, we recommend that you deposit your laboratory protocols in protocols.io to enhance the reproducibility of your results. Protocols.io assigns your protocol its own identifier (DOI) so that it can be cited independently in the future

• As the study did not involve any laboratory process, no protocols were deposited in protocols.io

• There are no changes to the financial disclosure

Please ensure that your manuscript meets PLOS ONE's style requirements, including those for file naming

While revising your submission, please upload your figure files to the Preflight Analysis and Conversion Engine (PACE) digital diagnostic tool, https://pacev2.apexcovantage.com/. PACE helps ensure that figures meet PLOS requirements.

• We have reformatted the manuscript according to the above style guidelines. 

• We have also amended, resized, and resubmitted all figures using PACE tools to ensure compliance with requirements.

Reviewers' comments:

Reviewer's Responses to Questions

Comments to the Author

1. Is the manuscript technically sound, and do the data support the conclusions?

The manuscript must describe a technically sound piece of scientific research with data that supports the conclusions.

Experiments must have been conducted rigorously, with appropriate controls, replication, and sample sizes. The conclusions must be drawn appropriately based on the data presented.

Reviewer #1: Partly

Reviewer #2: Yes

• We appreciate the feedback on the study design and data. In response to your concerns on the technical aspects of the costing, we have substantially detailed the methods and the rationale behind them. The specific feedbacks are answered according to the highlighted comments by the reviewers in the following sections. [page 4, line 131-132; page 5, line 144-148; page 6, line 185-192; page7, line 208-214; page 8, line 230-255].

• We have also provided additional details on the sample size calculation, together with the supporting evidence for the selection of method and data [page 5, 144-148]. 

2. Has the statistical analysis been performed appropriately and rigorously?

Reviewer #1: Yes

Reviewer #2: Yes

• We appreciate the reviewers' acknowledgment of the appropriateness and validity of our statistical analysis

3. Have the authors made all data underlying the findings in their manuscript fully available?

Reviewer #1: No

Reviewer #2: Yes

• We have made the full dataset available in Harvard Dateverse, which is accessible from https://doi.org/10.7910/DVN/2SGZ9T. 

• Annual cost estimates and differences calculated using the IPW method is reported as part of the Supplementary material in S3 Table 

4. Is the manuscript presented in an intelligible fashion and written in standard English?

Reviewer #1: Yes

Reviewer #2: Yes

• We appreciate the positive comments on the standard of the written language

5. Review Comments to the Author

Reviewer #1: The article attempts to determine the costs of treating oral potentially malignant disorders (OPMD) and oral cancer in Malaysia from a healthcare provider’s perspective. The authors have aimed to fill an important missing gap in the current literature on the cost of OPMD and oral cancer in Malaysia and also to provide policy recommendation on the need for early detection and prevention.

Besides trying to fill an important gap in literature, the study includes both East and West Malaysia resources use which is one of the key strengths of the study as well as the comprehensiveness of the study.

However, the study has several limitations which may need to be addressed:

Time of study period is short (6 months):

To do a comprehensive costing, time-period should be longer to capture follow-up treatment of cancer and resources use. Explain why this time period is used and whether the study could be extended for a longer time period.

• We appreciate the reviewer’s view in recognizing the gap in knowledge in establishing the public healthcare cost to manage OPMD and oral cancer in the region. We agree with the reviewer on the importance of conducting prospective longitudinal studies to gather comprehensive data on follow-up and services consumed. However, such a design consumes a huge amount of resources and time due to variable local incidence rates, unavailability of centralized medical records, complex and diverse interdisciplinary management at various units/institutions. Thus to overcome these challenges, we have decided to collect the data retrospectively based on patients being followed up in the specialist clinics. This allows us to capture all treatment and management costs incurred based on clinical events recorded in medical records [page 8, line 237-245]. 

• Although the study period was only 6 months, the average follow-up duration (post-1-year of diagnosis) obtained from our retrospective case note abstraction ranged from 42 to 61 months [Table 1]. These data were still able to mirror the resources consumed in the maintenance phase. Interestingly, this approach also better represents the spectrum of the reality of cost incurred in public hospitals as it consists of a myriad of patient case lengths and conditions. Thus the values generated gave a good approximation of the actual cost borne by the Ministry of Health at any specific time [page 17, line 441-451].

Background

Provide background of health system in Malaysia and whether all the costs determined in the analysis are covered by the government. Are there any out-of-pocket costs or costs from private insurers that are not included?

• Thank you for pointing out this critical point. We have included a background on the healthcare system in Malaysia in the Introduction section [page 2, line 70-77].

• The primary aim of the study is to provide critical information on OPMD and oral cancer management from a public healthcare provider's perspective. This is because management in public health care facilities is heavily subsidized and predominantly financed through taxation and revenues from the federal government. Furthermore, the recent implementation of financial incentive schemes for cancer patients, in addition to increasing disease demands and changing demographics will primarily impact the financial allocations, necessitating information for prioritization of resources [page 3, line 90-92]. Thus the study incorporated neither out-of-pocket costs nor private insurers. 

• We have added the term ‘public healthcare provider’ to the aim to clarify the focus of the study [page 3, line 105].

Costing seems incomplete.

Since the study did not fully include one of the key drivers of costs based on external institutions data (hospitalization and utilization of chemo) and instead relied on expert opinion (perhaps one 1 expert), the costing is not fully accurate. To do a full costing, attempts should be made to get data from the National Cancer Institute and other external institutes to obtain the resource use of hospitalizations and chemo treatment.

Provide more details on the constitution of the inter-disciplinary team – which institutes etc.? Any reason a hospital pharmacist was not involved who might be able to provide more cost details and treatment pathways?

• We appreciate the motivation for these comments. We would like to clarify that only the hospitalization for chemotherapy and radiotherapy was estimated based on experts' opinions. Hospitalization for other inpatient care (such as surgical interventions) was still abstracted precisely from medical records. 

• Summary of the treatment regimen for chemotherapy and radiotherapy plan was obtained from National Cancer Institute (NCI) (for patients managed in HTAR) and respective oncology unit in HUS. However, the total duration of hospitalization was often not included in the summary report. Thus to ensure standardization, hospitalization for each patient was estimated based on their individual regimen/plan following routine and standard practice. The minimum period of hospitalization was obtained from expert opinions (oncologists, radiotherapists, and two pharmacists involved in planning chemotherapy regimens in NCI and HUS) [page 7, line 208-214]. As suggested, hospital pharmacists were closely involved in providing cost details of drugs, additives, and standard investigative tests based on adopted regimens. We have expanded the Valuation and Sources section to reflect this [page 8, line 208-220].

• Nevertheless, as highlighted by the reviewer, this may underestimate the total number of days of hospitalization for chemotherapy/radiotherapy from factors such as complications and additional costs involved in treating them. We have added this limitation to our Discussion [page 18, line 475-480].

More information required on the types of chemo drugs that were considered as part of the costing.

• The type of chemotherapy regimens adopted varied greatly from disease status, patient response and tolerance, to availability of drugs. We have added the most common regimens used (consisting of Cisplatin, Carboplatin, 5-Fluorouracil, Docetaxel, and Gemcitabine) in the Result section as it may help the readers to have a clearer picture of the cost components as suggested by the reviewer [page 12, line 333-335].

Also needs to show the variation of costs for different patients– may need some form of probability distribution function to show the variation in costs is recommended.

• Upon discussion with the team, it was decided that reporting the patient cost by treatment type and stratified by diagnosis was the most informative for the policy-makers. We have incorporated the distribution of values based on 95% confidence interval in Table 3, in addition to the mean (SD), minimal and maximum possible values per patient [page 13, Table 3].

• Furthermore, all reported mean in the result section was updated with the inclusion of 95% confidence interval for the estimate [page 11, line 305-312; page 12, line 331-335; page 13, line 341-342]. For IPW estimates, due to space constraints, the 95% confidence interval values are reported in S3.

Consider inclusion of process map that shows the journey of the patient through the healthcare system

• We have taken this suggestion and included ‘Fig 2: Simplified clinical pathways of the patient journey in a public healthcare facility’ to describe the clinical journey of patients. We have chosen to represent the clinical pathway for the treatment of cancer in the buccal mucosa, the floor of mouth, tongue, alveolar ridge, hard palate based on NNCN Guideline v1.2019 as this was the most common in our sample. Details on the incorporated components are shown in Fig 1 in detail.

The title only indicates oral cancer when the study includes both oral cancer and OPMD costing.

• Thank you for highlighting the improvement. To better frame the main aim of our paper, we have amended the title to “Provider cost of treating oral potentially malignant disorders and oral cancer in Malaysian public hospitals”

Method

More rationale provided on the IPW method and why the authors think this costing method is the most appropriate for undertaking this study and how the costs were adjusted for follow-up and survival. Have the authors considered other methods and if so, why are those methods excluded from the study? How does the study handle uncertainty in costs and the skewed nature of healthcare costs?

• Thank you for highlighting the importance of describing the rationale for the selection of the IPW method. Primarily the decision for the selection of method was attributable to the study design and sampling method we had adopted. These introduced large variation in case length and more importantly an ‘induced informative censoring’. As a result, traditional methods for handling censored survival data, such as the Kaplan–Meier estimator or Cox proportional hazards regression model, may no longer be valid for analyzing the data. While several methods such as Bang and Tsiatis (BT) estimator, Bang and Tsiatis partitioned version (BTP) and Zhao and Tian (2001) (ZT) estimator are available, the BT estimator was selected because our dataset was collected and recorded in the format of single-record-per-subject survival data, rather than multiple-record-per-subject survival data needed for ZT estimator. We have added a summarised rationale for the selection of the method [page 8, line 237-245].

• While cost data often do not conform to parametric assumptions, the estimates were described in means, following the good research practice guidelines for cost analysis by the Professional Society for Health Economics and Outcomes Research (ISPOR). This was to ensure pragmatism in guiding decisions [page 9, 269-272]. 

• However, to ensure the robustness of the analysis, comparisons between diagnosis groups were still conducted using non-parametric analysis such as the Kruskal-Wallis test [page 9, 263-264]. To capture the uncertainties around the mean estimates, we have added and reported the confidence interval in the report mean values, Table 3 and S3 Table.

What constitute maintenance costs?

• In the study, the initial and maintenance phase cost consists of all four components as shown in Fig 1- namely outpatient, biopsy, investigation, and inpatient care. Cost data were split based on the time from diagnosis rather than by treatment component. This was to ensure all factors and components of healthcare utilization were captured within the reported time frame [page 6, 181-185; page 8, line 225-229]. 

The study did not seem to include any administrative costs in the total costs (e.g. utilities, personnel, surgical overhead costs). These may need to be determined or approximated so that costs provided are not under-estimated.

• The ‘Full-Paying Non-Citizen’ tariff in the Medical Fees Order (Cost of Services) 2014 represents the unsubsidized cost in the public healthcare facilities. It was not possible to obtain information on how these values were calculated. However, it was based on surveys on private fees, consultations with heads of specialty, and available cost information. A crude comparison of selected dental procedures fees generated from a local cost analysis study validates the cost to be inclusive of capital, operation, maintenance, administrative, and consumables used. We have added this information to the Valuation and sources section [page 6, line 185-192]. 

• We have nevertheless highlighted that applying hospital tariffs may underestimate expenditures from fixed cost components such as shared facilities and other indirect health consequences besides elements like administrative costs in the Discussion section [page 18, line 484-486]. 

Inclusion of sensitivity analysis to show variation in costs

• We appreciate the suggestion from the reviewer on the inclusion of sensitivity analysis to show variation in cost. We chose to demonstrate the variation in cost by using the total treatment cost stratified by type of therapy in Table 3. The min and max values give a close approximation of treatment cost ranges based on the actual case. We acknowledge that our estimations for the hospitalization and utilization for chemotherapy and radiotherapy had introduced uncertainties in the values generated, and may underestimate actual therapy cost. However, a sensitivity analysis is difficult to be conducted for both these components because the input parameters vary drastically according to each selected regimen and associated risks. This may cause our reported values to lose their intuitiveness and ease of interpretation. 

Suggest to change the word ‘manpower’ to personnel, workforce or professional services etc

• As suggested, we have substituted “manpower” to “personnel” to be more apt with our study context [page 1, line 50; page 12, line 324].

Reviewer #2: Authors have attempted to investigate important area of research. Economic cost assessments are very important to eye open the policy makers. Following comments to improve the manuscript for publication.

1 Generally, costing studies describes direct and indirect cost. However, this study focuses only on direct health care cost which is part of the costing analysis.

• We appreciate the positive comments from Reviewer #2. We have focused on the direct health care cost (as clarified in the Introduction and earlier responses) because under the current public healthcare system, clinical services are heavily subsidized and predominantly financed by the federal government. Moreover, additional subsidization and financial incentive for cancer patients may further add to the financial burden in ensuring universal health coverage. Thus direct public provider costs are primarily the most critical information for prioritization of resources and consideration for investment in preventive care by the ministry. Thus the main objective of the study was to estimate and characterize the provider cost (direct) rather than the indirect costs [page 3, line 103-105]. 

2. Term “oral cancer”, generally exclude salivary gland tumor, what is the specific reason to include the ICD C7 and C8

• Thank you for pointing out this important subject. While we have initially adopted broad inclusion criteria for ‘oral cancer’, consisting of ICD 00 to 08, all sampling, reporting, and analysis excluded ICD 07 and 08. To avoid confusion and to be in line with the study, we have changed the reported term to consist of ICD 00 to 06 [page 5, line 157].

3 Is the follow up visit cost taken into considerations.

• Yes, all follow-up visit costs were taken into consideration in the annual cost estimate at both phases. This consisted of the specialist clinic visit, dental examinations or any other investigations conducted and medications prescribed [page 8, line 223-229]. 

 4. Generally OPMD patients are treated in the Out Patient Department, however, considerable amount money spent on inpatient segment in table 2.

• Table 2 reports the initial phase (first year) expenditures. Based on our sample, this higher inpatient cost in the first year was contributed by a high surgical intervention cost of 16 patients. As we adopted a stage-stratified convenience sampling in the specialist clinics, there was a possibility of more severe cases of OPMD (requiring surgical interventions) to be encountered and recruited. It also elucidates a potential cost driver in the management of late diagnosis of OPMD which may have not been captured in routine standardized samples [page 13, Table 3]. 

5. In cost calculation, valuation was based on the “full paying patient”, according to my understanding, is it subsidized valuation or profit based valuation. It is not clear to the reader.

• Although the Ministry of Health (MoH) reports the overall expenditures to maintain the public healthcare system, comprehensive cost information per service is currently not fully characterized. The legislated Fees Act 1951 is based on a subsidized valuation rather than profit-based. The information was added to the Valuation and sources section [page 6, 186-187]. 

• The non-citizen rate was applied as a cost proxy as it best represents unsubsidized charges for clinical services in public healthcare facilities. To avoid confusion, we have also changed the ‘full-paying patient’ tariff to an unsubsidized ‘full-paying non-citizen’ tariff [page 6, 185].

---

## [Decision Letter · Decision Letter 1]

3 May 2021

Provider cost of treating oral potentially malignant disorders and oral cancer in Malaysian public hospitals

PONE-D-21-05143R1

Dear Dr. SHAFIE,

We’re pleased to inform you that your manuscript has been judged scientifically suitable for publication and will be formally accepted for publication once it meets all outstanding technical requirements.

Kind regards,

Susan Horton

Academic Editor

PLOS ONE

Reviewer #1: All comments have been addressed

---

## [Editor Report · Acceptance letter]

5 May 2021

PONE-D-21-05143R1 

Provider cost of treating oral potentially malignant disorders and oral cancer in Malaysian public hospitals 

Dear Dr. Shafie:

I'm pleased to inform you that your manuscript has been deemed suitable for publication in PLOS ONE. Congratulations! Your manuscript is now with our production department. 

Kind regards, 

on behalf of

Dr. Susan Horton 

Academic Editor

PLOS ONE